# UAV approaches for improved mapping of vegetation cover and estimation of carbon storage of small saltmarshes: examples from Loch Fleet, northeast Scotland

William Hiles[1], Lucy C. Miller[1], Craig Smeaton[1] and William E.N. Austin[1,2]

[1]School of Geography and Sustainable Development, Irvine Building, North Street, University of St Andrews, Fife, KY16 9AL, United Kingdom
[2]Scottish Association of Marine Science, Scottish Marine Institute, Oban, Argyll, PA37 1QA, United Kingdom

Correspondence to: William Hiles (wh34@st-andrews.ac.uk)

**Abstract.** Saltmarsh environments are recognised as key components of many biophysical and biochemical processes at the local and global scale. Accurately mapping these environments, and understanding how they are changing over time, is crucial for better understanding these systems. However, traditional surveying techniques are time-consuming and are inadequate for understanding how these dynamic systems may be changing temporally and spatially. The development of Uncrewed Aerial Vehicle (UAV) technology presents an opportunity for efficiently mapping saltmarsh extent. Here we develop a methodology which combines field vegetation surveys with multispectral UAV data collected at two scales to estimate saltmarsh area and organic carbon storage at three saltmarshes in Loch Fleet (Scotland). We find that the Normalised Difference Vegetation Index (NDVI) values for surveyed saltmarsh vegetation communities, in combination with local tidal data, can be used to reliably estimate saltmarsh area. Using these area estimates, together with known plant community and soil organic carbon relationships, saltmarsh soil organic carbon storage is modelled. Based on our most reliable UAV-derived saltmarsh area estimates, we find that organic carbon storage is 15-20% lower than previous area estimates would indicate. The methodology presented here potentially provides a cheap, affordable, and rapid method for saltmarsh mapping which could be implemented more widely to test and refine existing estimates of saltmarsh extent and is particularly well-suited to the mapping of small areas of saltmarsh environments.

## 1. Introduction

Saltmarshes have been identified as significant hotspots for the storage and burial of organic carbon (OC) at the land-ocean interface (Duarte et al., 2005; Mcleod et al., 2011), and the OC stored within saltmarshes is increasingly recognised as playing a role in climate mitigation (Nelleman et al., 2009). However, these habitats are under increasing pressure from sea level rise, climate change, and anthropogenic disturbance (Pendleton et al., 2012). It has been estimated that 0.28% of global saltmarsh area is lost annually, with 719 km$^2$ lost between 2000 and 2019 (Campbell et al., 2022). Crucial to understanding the quantity of OC stored in saltmarshes and their climate mitigation potential is understanding the areal extent of the habitat and how it changes through time, yet until recently the area of global saltmarsh has been poorly constrained with estimates ranging

between 22,000 km$^2$ and 400,000 km$^2$ (Mcleod et al., 2011). This has resulted in global saltmarsh OC stock estimates ranging between 0.4 Gt OC and 6.5 Gt OC (Duarte et al., 2013) with a further 10.2 Mt OC to 44.6 Mt OC being buried annually (Chmura et al., 2013; Ouyang and Lee, 2014). The broad ranges of global saltmarsh OC stocks and burial rates highlights that

something as straightforward as accurately estimating saltmarsh area is crucial to understanding the much more complex biogeochemical cycles typical of these environments. These include the need to precisely quantify saltmarsh OC dynamics in order to assist decision makers in implementing climate change mitigation measures. Human activities impact these natural carbon sinks through land use, land-use change, and forestry (LULUCF). As a result, exchange of $CO_2$ between the biosphere and atmosphere can be altered by management interventions that can contribute to climate mitigation.

More recently estimates of global saltmarsh environment have been revised downwards to 54,951 km$^2$ (Mcowens et al., 2017), which is in the lower end of previous estimates (Mcleod et al., 2011). The revised global saltmarsh areal extent is based on habitat data provided by individual nations. However, the identification, mapping, and long-term monitoring of saltmarshes is a labour-intensive and time-consuming process; as a result many nations have an incomplete understanding of the area of their saltmarsh environments (Mcowen et al., 2017) or their estimates are outdated as the surveys were undertaken some time

ago and potentially no longer accurately represent these highly dynamic systems that are constantly eroding and accreting (Ladd et al., 2019). Consequently, methods for the rapid identification and long-term monitoring of saltmarshes that reduce time and cost commitments are crucial for maintaining accurate estimates of saltmarsh extent and understanding saltmarsh OC storage.

Saltmarshes are typically identified using two methods. Firstly, they may be classified according to tidal ranges. The maximum

elevational limit of saltmarsh vegetation environments is the high astronomical tide (HAT), where periodic inundation at the extremes of the tidal range still creates saline environments (Foster et al., 2013). The lowest limit of saltmarsh environments, on the other hand, appears to be located at an altitude where the frequency of tidal inundation starts to decrease from its maximum value, which corresponds approximately with the mean high water neaps (MHWN) (Pye and French, 1993; Balke et al., 2016; Ladd et al., 2019). Due to the influence of tides and the exposure to saline water saltmarshes typically develop

characteristic vegetation communities dominated by halophytes (Adam, 1978; Burd, 1989); therefore, a second approach to classifying saltmarsh environments is to map vegetation communities. This approach to vegetation mapping requires highly specialised surveying and it very labour-intensive. More recently, remote sensing methods have proven to be successful at identifying and monitoring intertidal environments (Murray et al., 2019; 2022; Bunting et al., 2022) including saltmarsh vegetation (Campbell and Wang, 2019; Blount et al., 2022; Stückemann and Waske, 2022). While natural colour satellite

imagery may help to delineate saltmarsh vegetation in coastal settings, recent advances in multispectral data availability has allowed saltmarsh identification using key vegetation indices, such as the Normalised Difference Vegetation Index (NDVI) (Tucker, 1979). The NDVI uses the relative absorption of the visible red and reflection of the near-infrared (NIR) wavelengths to estimate plant health and can potentially be used as a tool for mapping both the extent and zonation of saltmarsh vegetation (Sun et al., 2016; Laengner et al., 2019). NDVI data has long been available from satellite imagery. However, the resolution

of satellite data reduces its effectiveness when attempting to resolve relatively small features such as smaller saltmarsh areas.

The advent and development of uncrewed aerial vehicles (UAVs) presents increasing opportunities for rapid, affordable, and repeatable mapping of saltmarsh extent. Unlike satellites, UAVs may be used in a highly targeted way to collect specific data and, due to the low elevations of the flights, issues that arise in satellite data (such as cloud cover and atmospheric modulation of signals) are not a significant concern (e.g. van Leeuwen et al., 2006; Nagol et al., 2009). Over the last decade, the capabilities of UAVs and post-processing software have increased significantly, potentially revolutionizing saltmarsh mapping (e.g. Doughty and Cavanaugh, 2019; Pinton et al., 2021; Chen et al., 2022) and providing a platform to allow dynamic OC stock assessments which can keep pace with the natural or anthropogenically driven changes affecting these habitats.

In this study we present data from three saltmarshes in northeast Scotland, combining vegetation surveys, tidal data and UAV imagery generated at two spatial resolutions. Here we: **i)** test and develop a straightforward method which uses different types and resolutions of UAV data, in conjunction with independent tidal data, to map saltmarsh environments based on vegetation multispectral indices; **ii)** compare traditional methods of estimating saltmarsh extent with estimates derived from UAV data; iii) use estimates of carbon storage as a case study to demonstrate the importance of accurately mapping saltmarsh areal extent.

## 2. Methods

### 2.1 Study area

There are 240 mapped saltmarshes in Scotland which occupy a combined area of 58.68 km$^2$ (Haynes, 2016), representing approximately 13% of the United Kingdom's saltmarsh environments. The coastal geomorphology of Scotland results in a large number of small saltmarshes averaging 0.25 km$^2$, with only nine marshes greater than 1 km$^2$ in size located in Scotland. These saltmarshes are estimated to store $0.94 \pm 0.26$ Mt of OC, with a further 4,385 tonnes of OC buried annually (Miller et al., 2023; Smeaton et al. ,2023). Loch Fleet is a small (6.94 km$^2$) spit-enclosed macrotidal bay situated in the northeast of Scotland (Fig.1A). The large tidal range results in an inter-tidal area which encompasses 75% of the bay, 0.2 km$^2$ of which is occupied by saltmarsh systems (Fig.1B). These saltmarshes were last surveyed in 2011 (Haynes, 2016) following the National Vegetation Classification (NVC) scheme approach (Rodwell, 2000). The saltmarsh in Loch Fleet were classified as embayment marshes ranging between 0.01 km$^2$ (Creag Bheag) and 0.08 km$^2$ (Balbair) in size. This survey identified five saltmarshes in Loch Fleet. The mapping combined field surveys with aerial photography, and vegetation communities were mapped to generate GIS polygons by combining the field and remote data. The vegetation at these marshes is dominated by four communities which occur in varying proportions: *Salicornia/Suaeda, Puccinellia, Puccinellia/Festuca*, and *Juncus gerardii* which is analogous with eastern Scotland saltmarshes (Adam, 1978; Burd, 1989). Additionally, it is estimated that 1,790.6 tonnes of OC is held within the superficial (top 10 cm) soils of the five saltmarshes within Loch Fleet (Smeaton et al., 2022). The previous surveys found it difficult to map fragmented edges of saltmarshes due to the resolution of the aerial imagery, and to account for pans and creeks within the saltmarsh. They also found it difficult to constrain gradational landward constraints on saltmarshes, particularly where saline influences were not markedly visible in vegetation communities (Haynes, 2016). We suggest that high-resolution UAV data may help to better constrain these edges.

Of the five mapped saltmarshes, three were selected for this study (Fig.1B): Skelbo, Creag Bheag and Cambusmore Lodge. These were selected as they represent a variety of saltmarsh sizes, positions relative to the open ocean, and different

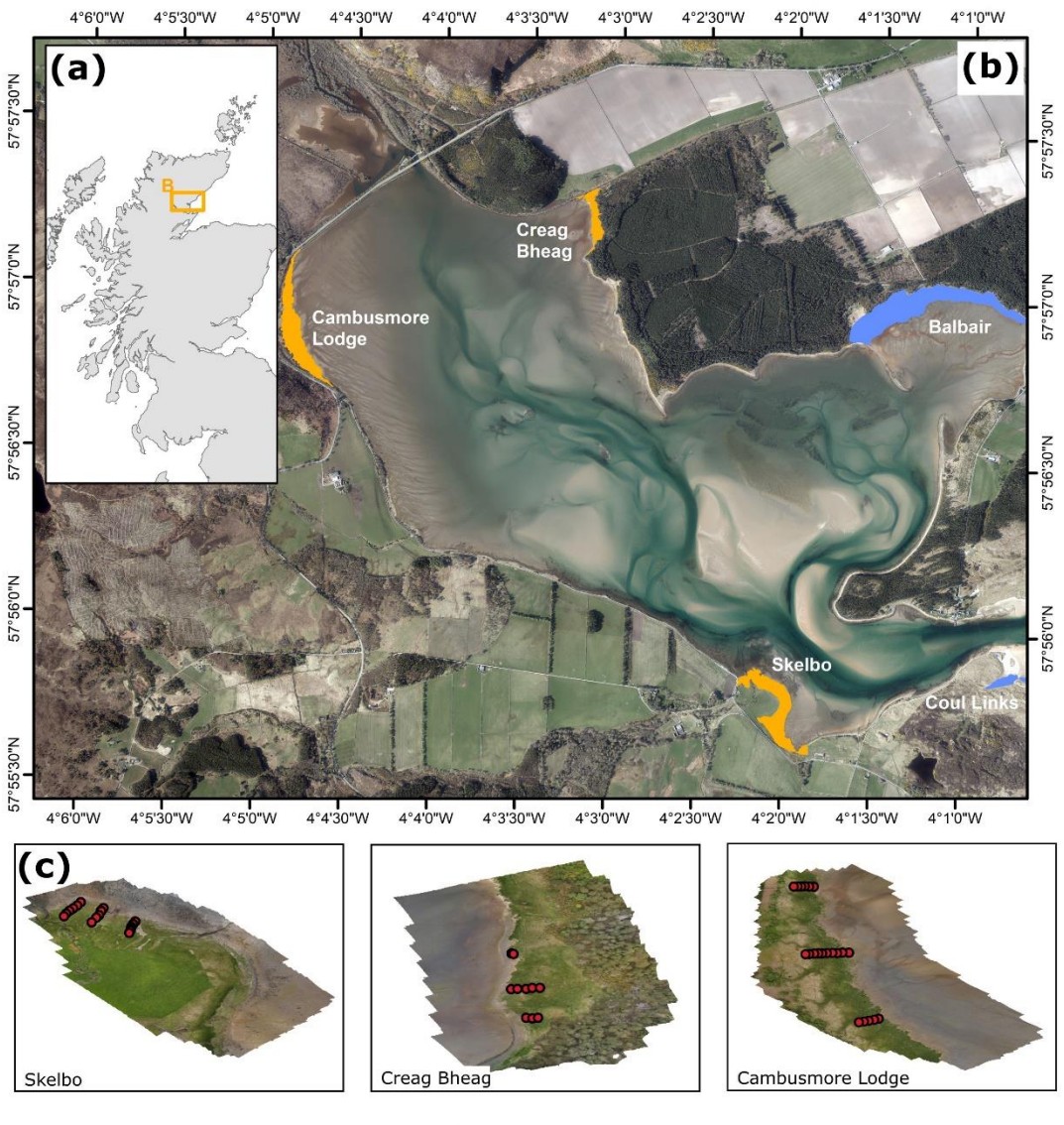

● Quadrat Location

**Figure 1. Saltmarshes of Loch Fleet. (a) Location of Loch Fleet within the United Kingdom, and locations of tidal gauges used in this study. (b) The location of the saltmarshes in Loch Fleet with the sites selected from this study highlighted (orange) and other Loch Fleet saltmarshes (blue). (c) UAV-generated orthomosaics created through this study, with quadrat sampling locations. Saltmarsh**
**mapping data extracted from Haynes (2016). Basemap © Ordnance Survey via © Getmapping Plc.**

surrounding vegetation types, while retaining similar climatic and oceanographic conditions. Together the three sites represent 50% of the current mapped areal extent of saltmarsh in Loch Fleet and are estimated to store 907.9 tonnes of OC in the top 10 cm of their soils (Smeaton et al., 2022).

### 2.2 Study sites

#### 2.2.1 Skelbo

Skelbo is a small saltmarsh estimated to cover 0.04 km$^2$ (Haynes, 2016). The saltmarsh edge primarily faces north and east and is the closest of our sites to the outlet into the North Sea, which is approximately 2 km from the site. A small stream (Skelbo Burn) runs through the saltmarsh, with birch trees growing on its banks. Parts of Skelbo are managed; a fence passes through the saltmarsh, and cattle periodically graze inland of the fence.

#### 2.2.2 Creag Bheag

Creag Bheag is the smallest saltmarsh investigated in this study and is estimated to cover 0.01 km$^2$ (Haynes, 2016). The saltmarsh is in the northeast corner of Loch Fleet and faces east, in the opposite direction to the outlet into the North Sea. Creag Bheag is situated on the western edge of a pine-dominated forest, and is relatively undisturbed (with the exception of a rail track just to the north).

#### 2.2.3 Cambusmore Lodge

Cambusmore Lodge is the largest saltmarsh investigated in this study and is estimated to cover 0.05 km$^2$ (Haynes, 2016). The saltmarsh is located at the westernmost edge of Loch Fleet with the edge to the east. The outlet to the North Sea is approximately 5 km to the southeast. Cambusmore Lodge is bordered by a small birch woodland on its western edge, and a main road is just 0.2 km from the saltmarsh edge at its greatest distance.

### 2.3 Site surveys

In June 2022, at each of our selected saltmarshes, three transects from the high marsh edge to the saltmarsh edge were established (Fig. 1). Along each transect several 1 m$^2$ quadrats were placed approximately every 10 m or at abrupt changes in vegetation. The location and elevation of each quadrat (Skelbo, $n=18$; Creag Bheag, $n=10$; Cambusmore Lodge, $n=21$; Fig. 1) was recorded using a Juniper Geode GSN3 dGPS. Latitude, longitude, and elevation were recorded with an accuracy of 2 cm. At each quadrat, the vegetation plant species were identified, and their percentage coverage was estimated by eye following standard NVC methodology (Rodwell, 2000) as used by Haynes. (2016). Additionally, the mean (n = 5) and maximum (n = 1) vegetation heights were determined with the quadrat following Stewart et al. (2001).

### 2.4 UAV data collection

At each site, several UAV mapping flights were conducted using a DJI Phantom P4 multispectral UAV. All flights were conducted using the DJI RTK2 unit, which increases hover accuracy to 10 cm horizontally and vertically. In addition to RGB

images, this camera model captures blue ($\lambda = 450 \pm 16$ nm), green ($\lambda = 560 \pm 16$ nm), red ($\lambda = 650 \pm 16$ nm), red edge ($\lambda = 730 \pm 16$ nm), and near infrared ($\lambda = 840 \pm 26$ nm) images with a resolution of 2.12 MP. The spatial resolution of the images obtained by each sensor can be calculated as follows:

$$R = \frac{H}{18.9} \qquad\qquad\qquad\qquad\qquad\qquad\qquad (eq.1)$$

Where R is the image resolution (cm pix$^{-1}$) and H is the relative height of the UAV above the ground (metres). The DJI Phantom
P4 Multispectral uses a sunlight sensor positioned on the top of the UAV body to allow internal correction for changes in light intensity.

The UAV was used to generate images at two resolutions. Firstly, the UAV was flown at a height of 2 m above each quadrat in order to generate a high-resolution (approximately 1 mm pix$^{-1}$) image of each surveyed vegetation community. At least three images were captured of each quadrat. Secondly, the UAV was flown at a height of 75 m above the ground over an area
designed to capture a large proportion of each saltmarsh area. These flights were performed to allow the generation of large saltmarsh maps at a resolution of approximately 4 cm pix$^{-1}$. Flights were undertaken with a horizontal overlap of 85% and a vertical overlap of 70% to provide a sufficient number of tie-points between adjacent images. Five ground control points (GCPs) were placed across the surface of the marsh and positioned using the dGPS during the 75 m flights to refine the geospatial accuracy of the orthomosaics.

**2.5 Post-processing**

**2.5.1 Vegetation classification**

Post-processing followed the processes outlined in Fig. 2. Vegetation data was processed into the NVC scheme (Rodwell, 2000) using Modular Analysis of Vegetation Information System (MAVIS). This method uses a United Kingdom reference database for vegetation communities and, using multivariate statistical methods, assigns survey data to an established
community based on the community composition (Table 1). Some communities may be classified as a mosaic, being comprised of one or more sub-community, but where this occurs the dominant community is used. Where the prefix "SM" occurs in front of a numerical value, this denotes a saltmarsh vegetation community classified using the NVC approach.

**Table 1: NVC Community classifications for vegetation communities identified at Loch Fleet saltmarshes. Community data from**
**Haynes (2016). Asterix (*) marks communities identified in the Haynes (2016) survey but not in the new survey presented here.**

| NVC Classification | Community | Saltmarsh Zone | Saltmarshes |
|---|---|---|---|
| SM8 | Annual *Salicornia* saltmarsh | Pioneer saltmarsh | S |

| SM10 | *Puccinellia maritima*, annual *Salicornia* species and *Suaeda maritima* | Transitional low-marsh | S* |
|------|------|------|------|
| SM13a | *Puccinellia maritima* saltmarsh, *Puccinellia maritima* dominant sub-community | Low-mid marsh communities | S |
| SM13b | *Puccinellia maritima* saltmarsh, *Limonium vulgare-Armeria maritima* sub-community | Mid-upper marsh communities | CL, CB |
| SM13d | *Puccinellia maritima* saltmarsh, *Plantago maritima-Armeria maritima* sub-community | Mid-upper marsh communities | S |
| SM16a | *Festuca rubra* saltmarsh, *Pucinellia maritima* sub-community | Mid-upper marsh communities | CL |
| SM16b | *Festuca rubra* saltmarsh, *Juncus gerardii* dominated sub-community | Middle marsh | CB*, S* |
| SM16c | *Festuca rubra* saltmarsh, *Festuca rubra-Glaux maritima* sub-community | Mid-upper marsh communities | CL, CB, S |
| SM16e | *Festuca rubra* saltmarsh, *Leontodon autumnalis* sub-community | Mid-upper marsh communities | CL |
| SM18a | *Juncus maritimus* saltmarsh, *Plantago maritima* sub-community | Upper marsh | S* |
| SM19 | *Blysmus rufus* saltmarsh | Wet depressions and channels | S* |
| SM28 | *Elytrigia repens* saltmarsh | Drift-line | CL |

## 2.5.2 UAV post-processing

All UAV imagery was processed using Structure-from-Motion approaches (e.g. Kalacska et al., 2017) within Agisoft

Metashape v. 1.8.2. To generate the high-resolution quadrat images (1 mm pix$^{-1}$) all images of each quadrat were imported

and corrected for changes in solar irradiance at the time of imaging using data collected by the UAV's sunlight sensor. Images

were then aligned. To scale the images appropriately, each edge of the quadrat was manually calibrated to 1 m and the diagonal

was calibrated to 1.4142 m. A dense cloud, mesh, and texture were then generated. Finally, an orthomosaic with calculated

NDVI values for each pixel was generated within Agisoft Metashape using the formula:

$$NDVI = \frac{Band\ 5 - Band\ 3}{Band\ 5 + Band\ 3}$$    (eq.2)

where Band 5 is the NIR wavelength ($\lambda = 840 \pm 26$ nm) and Band 3 is the red wavelength ($\lambda = 650 \pm 16$ nm) (e.g. Carlson and

Ripley, 1997).

Orthomosaics for the mapped saltmarsh area (4 cm pix$^{-1}$) were similarly processed. Images obtained in the 75 m flight were

imported, corrected for light intensity using the UAV's internal sunlight sensor, and aligned. GCPs were manually identified

in the images and used to optimise the alignment. Three orthomosaics were generated for each site (Fig. 1, Fig. 3): an RGB

image, a digital surface model (DSM), and an NDVI image calculated using equation 2.

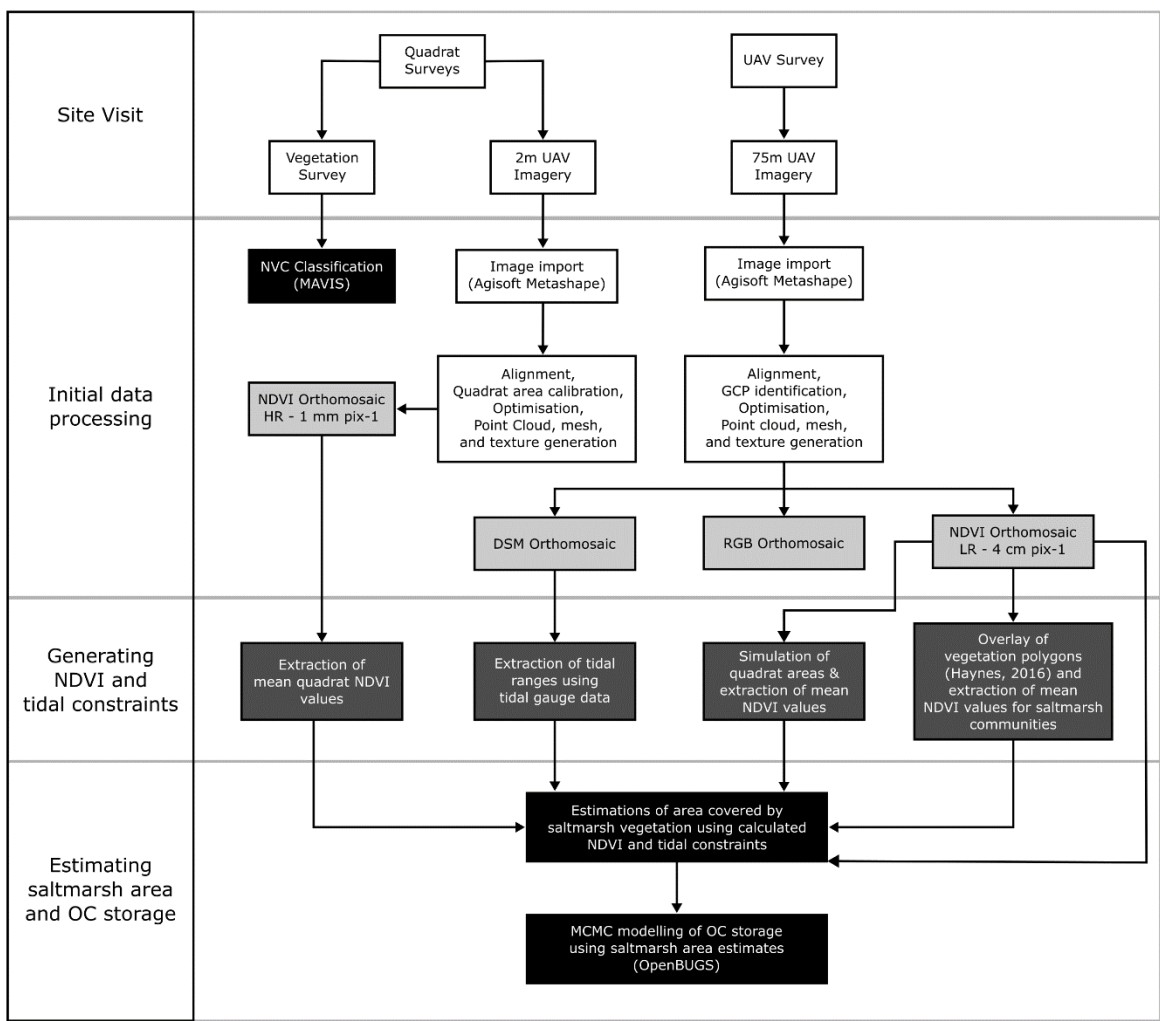

**Figure 2: Workflow schematic for this study. Light grey colouring represents initial map outputs. Dark grey colouring represents points where key data were extracted for use in additional analyses. Black colouring represents final data outputs.**


## 2.6 Classifying saltmarsh units using UAV data

Several approaches have been proposed to statistically classify vegetation communities using remote sensing data (e.g. Villoslada et al., 2020; Wolff et al., 2023). These methods are typically used to differentiate between vegetation communities,
and rely heavily on having high computational power for their machine learning algorithms. One of our aims in this paper is to test a method that relies less on high computational power and uses as few input variables as possible, allowing stakeholders such as government agencies and monitoring programmes to easily implement the method. We therefore restrict our input

variables to NDVI, the most common multispectral index applied to vegetation, and tidal data, to test whether these are adequate to classify saltmarsh against non-saltmarsh communities, with no attempt to differentiate the different communities that exist on UK saltmarshes.

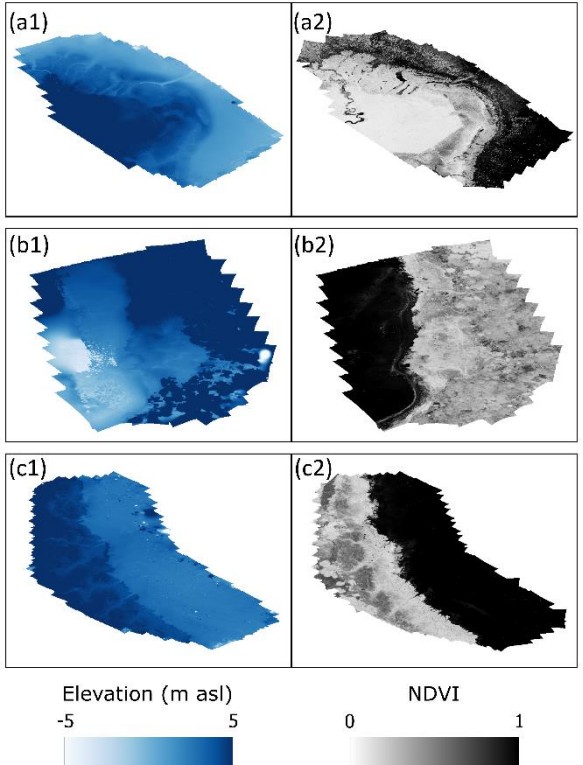

Elevation (m asl)
-5        5

NDVI
0        1

**Figure 3: 1) Digital Surface Models (DSMs) and 2) NDVI orthomosaics for (a) Skelbo, (b) Creag Bheag, and (c) Cambusmore Lodge**

### 2.6.1 Calculating saltmarsh area using tidal data

Saltmarsh area was first estimated using tidal data from the Wick (90 km, NE) and Aberdeen (140 km, SE) tidal gauges. Saltmarsh vegetation can colonise approximately around the Mean High Water Neaps (Balke et al., 2016), but the upper limit of saltmarsh communities occurs where the effects of salinity ends. One proposed threshold for this point is the High Astronomical Tide (HAT), the highest tidal level expected under a normal combination of astronomical conditions (e.g. Foster et al., 2013). For this study, we use the HAT to provide an upper elevation constraint to our maps. Using the 4 cm pix$^{-1}$ DSM orthomosaics generated using the UAV data and the tidal data from both Wick and Aberdeen gauges (Table 2; Fig. 1), the area expected to be inundated under each tidal condition was calculated.

**Table 2. Tidal ranges (above chart datum) from the Aberdeen and Wick tidal gauges used in this study. Data extracted from the National Tidal and Sea Level Facility.**

|  | Aberdeen | Wick |
|---|---|---|
| High Astronomical Tide (HAT) | 4.85m | 3.97m |
| Mean High Water Neaps (MHWN) | 3.46m | 2.78m |

### 2.6.2 Classifying vegetation using only NDVI data

Analysis of UAV-generated orthomosaics used QGIS v. 3.28. Three approaches were used to extract NDVI data from the UAV data. All three approaches use data from our new vegetation surveys or from the existing vegetation data from Haynes

(2016), in which vegetation communities have been classified using the NVC system. By combining these vegetation community data with high-resolution UAV data, relationships between vegetation communities and NDVI data can be developed at the millimetre to centimetre scale, providing a training dataset to classify the larger, unsurveyed vegetation communities across the mapped saltmarshes.

In the first of our approaches, the 1 mm pix$^{-1}$ images of each of our surveyed quadrats were imported into QGIS. Polygons

were manually created to cover the area inside the quadrat, removing the influence of the quadrat frame and removing vegetation outside of the surveyed quadrat. Average NDVI values were then extracted for the vegetation in each quadrat area. In our second approach, to explore the effect of image resolution on NDVI classifications the 4 cm pix$^{-1}$ orthomosaics were also imported into QGIS. dGPS location data for each quadrat was overlain on the orthomosaic and a 50 cm buffer was generated to simulate the 1 m$^2$ quadrat area. NDVI and elevation data were then extracted from each simulated quadrat area.

The third approach to extracting NDVI data utilised previously published data showing the estimated spatial distribution of vegetation communities based on the 2011 vegetation survey (Haynes, 2016). Polygons representing the area of vegetation communities (Fig. 4) were overlain on the NDVI and DSM orthomosaics, and the mean NDVI and elevation for each polygon was extracted. Where several polygons were assigned to the same vegetation community (e.g. where communities were non-contiguous and separated by another community) data from the polygons were combined.

Using these extracted NDVI values, several ranges of NDVI values associated with saltmarsh vegetation communities were calculated. For each image resolution (1 mm pix$^{-1}$ quadrats, 4 cm pix$^{-1}$ quadrats, and the Haynes (2016) polygons overlain on the 4 cm pix$^{-1}$ orthomosaic), and for each site, the saltmarsh vegetation communities with the highest and lowest mean NDVI values were identified. Using the NDVI data for these two communities, three ranges were calculated for use in further analyses: the range between the mean NDVI values of those two communities (hereafter the "Mean"), and the ranges between

the highest and lowest NDVI values with both one standard deviation ("1SD") and two standard deviations ("2SD") limits applied. At Skelbo, a significant part of the mudflat was covered by detrital material that was clearly not saltmarsh vegetation,

but yielded similar NDVI values to saltmarsh communities (Fig. 4). To account for this, data from the DSM was used to constrain the extraction to above 2 m asl.

In order to test whether NDVI data can effectively differentiate saltmarsh vegetation from bare mudflat, NDVI data was
extracted from mudflat areas at all three sites. Five points were placed on each mudflat in areas that were taken to represent range of visible NDVI values. At Skelbo and Creag Bheag some vegetation or algae was abundant on the mudflats, and points were intentionally placed on these areas to fully capture the possible non-saltmarsh NDVI signatures. At each point, a 1 m quadrat was simulated (as above), and the NDVI mean and standard deviation values were extracted. Additionally, a polygon was manually drawn around each mudflat using the RGB orthomosaic, being careful not to include any saltmarsh
environments, in order to estimate the average NDVI signature of the mudflats.

## 2.7 Estimating saltmarsh extent

Estimating saltmarsh extent combines the tidal data (Sect 2.5.1), the UAV-generated DSM, and the different NDVI determinations extracted from our field survey and the Haynes (2016) survey (Sect 2.5.2). Firstly, the area covered under three different tidal conditions were determined: the mapped area falling below the HAT of the Wick tidal gauge, falling below the
HAT of the Aberdeen tidal gauge, and the total area without applying any tidal constraint. Within each of these tidal constraints, the pixels with NDVI values falling within the established ranges (Sect 2.5.2) were extracted. The total area covered by pixels both falling within the tidal range (using the DSM) and the NDVI range (using the NDVI orthomosaics) is therefore taken to represent the saltmarsh environment determined by each combination of tide and NDVI range.

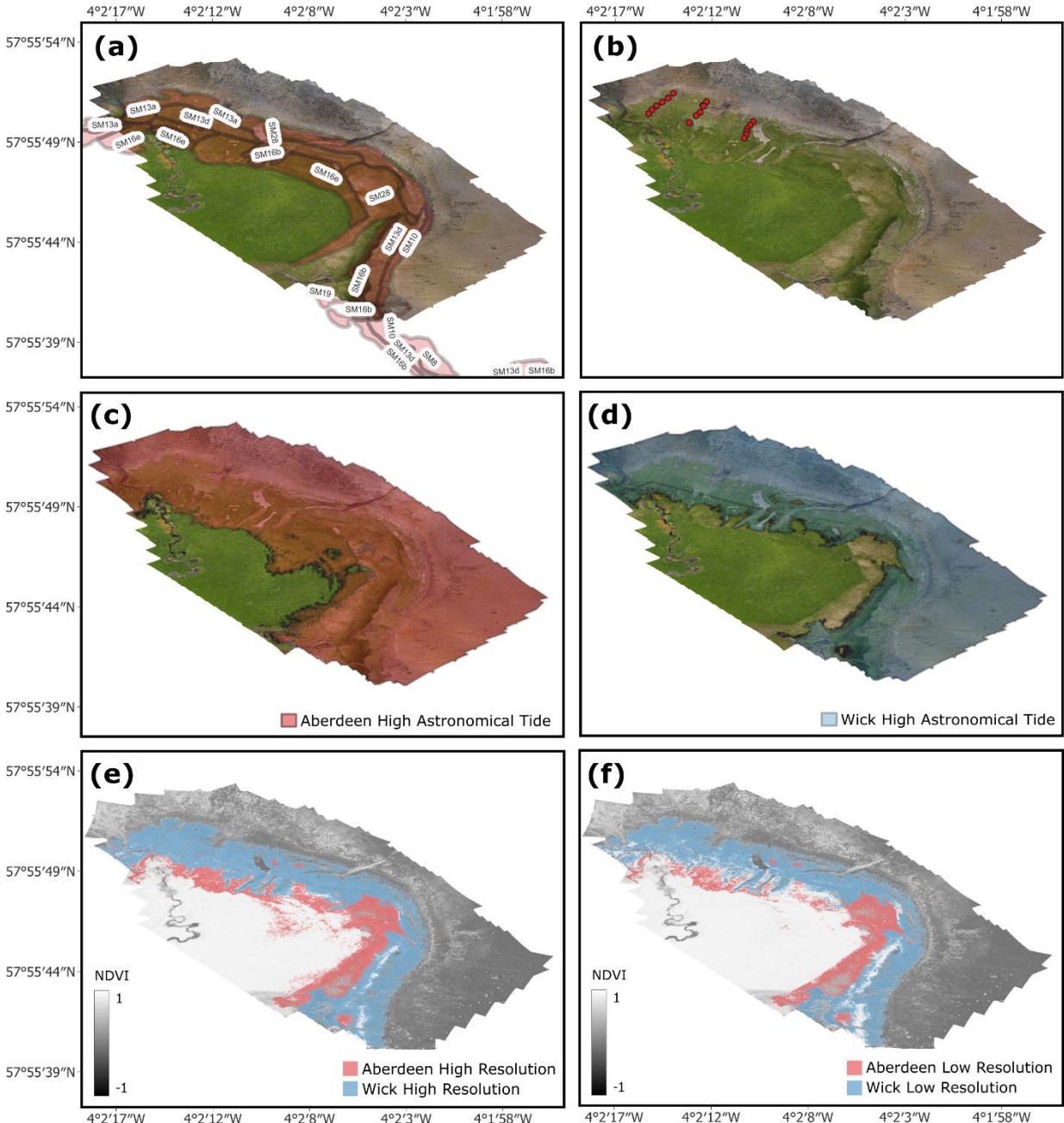

Figure 4: Comparison maps for estimating saltmarsh extent at Skelbo. (a) current mapped vegetation (Haynes, 2016); (b) and our quadrat sampling points; the tidal limits from the (c) Aberdeen and (d) Wick tidal gauges; saltmarsh extent estimates using NDVI classifications (Table 3) from the (e) high-resolution and (f) low-resolution UAV surveys, constrained by the Aberdeen and Wick tidal gauge data and using the 1 standard deviation range.

## 2.8 Estimating OC storage of saltmarshes

To estimate the OC stock of the saltmarshes, a surrogate OC storage value was calculated from values taken from two marshes in close proximity to Loch Fleet, Morrich More (13.2 km, SE) and Dornoch Point (8.44 km, S). The soils of Dornoch Point saltmarsh are estimated to store $19.5 \pm 7.4$ kg OC m$^{-2}$, while Morrich More stores $21.6 \pm 5.2$ kg OC m$^{-2}$ (Miller et al., 2023). The average OC storage value ($20.60 \pm 4.46$ kg OC m$^{-2}$) for these two saltmarshes was applied to the areal extent of each of the Loch Fleet marshes to calculate the OC stock (eq. 3).

$$\text{OC stock (kg)} = \text{OC storage (kg OC m}^{-2}) \times \text{Area (m}^2) \qquad\qquad \text{(eq. 3)}$$

The calculation steps were undertaken using a Markov Chain Monte Carlo (MCMC) approach using the OpenBUGS software package (Lunn et al., 2009). MCMC methods provide powerful and widely applicable algorithms for simulating data from sample distributions, allowing the incorporation of complex and multi-dimensional probability distributions (Andrieu et al., 2003). MCMC approaches take random values from the probability distribution curves of variables and use these to perform calculations. By repeatedly drawing random values from the distribution curves, summaries of thousands of calculations can be generated that robustly incorporate all possible results from the errors associated with the input variables. Here, we simulated OC stock calculations (eq. 3) 1,000,000 times assuming a normal distribution for OC storage ($20.60 \pm 4.46$ kg OC m$^{-2}$) and applying a $\pm 1\%$ error on our area estimates. We used 100,000 of these simulated outputs to summarise our estimates of Loch Fleet saltmarsh OC storage. This approach was used for each of the area estimates generated using the different methods in this study.

Haynes (2016) mapped saltmarsh vegetation but did not attempt to relate this to saltmarsh area or carbon storage. To provide a baseline against which to compare our OC storage estimates, this approach was also used on the saltmarsh area estimates generated from the existing Haynes (2016) vegetation community maps.

## 3. Results

### 3.1 Vegetation surveys

#### 3.1.1 Skelbo

Our vegetation survey at Skelbo identified four unique NVC communities representing the classic saltmarsh zonation: SM8 ($n = 1$; typical of pioneer saltmarsh), SM13a ($n = 7$; typical of low- to mid-marsh communities), and SM13d and SM16c ($n = 7$ and 3, respectively; typical of mid- to high-marsh communities). Using infield dGPS measurements, the vegetation follows an expected elevation gradient (Table 3).

In contrast, the previous survey of Skelbo identified eight different vegetation communities (Haynes, 2016). The only two communities in common between the surveys are SM13a and SM13d. The previous survey did not identify the pioneer community of SM8. However, it identified five communities not found in our new survey: SM10 (an additional low- to mid-

marsh community), and a suite of communities characteristic of mid- to upper-marsh environments (SM16b, SM16e, SM18a, and SM19).

### 3.1.2 Creag Bheag

Our vegetation survey at Creag Bheag identified only two vegetation communities: SM13a ($n = 3$) and SM16c ($n = 6$), both of which are characteristic of mid- to high-marsh zones. Elevation estimates of 3.68 ± 0.82 m asl and 2.68 ± 0.62 m asl, respectively, are consistent with this zonal setting (Table 3).

In contrast, the previous survey identified four vegetation communities at Creag Bheag (Haynes, 2016). The only community in common between the two surveys is SM16c. Unlike in our survey, SM13a was not previously identified. The previous survey additionally identified two mid- to upper-marsh communities (SM13b and SM16b), and SM28 (a driftline community).

### 3.1.3 Cambusmore Lodge

Our vegetation survey at Cambusmore Lodge identified five different saltmarsh communities. SM28, a drift-line community, was identified in a single quadrat. All other quadrats identified communities typical of mid- to upper-marsh environments: SM13b ($n = 2$), SM16a ($n = 4$), SM16c ($n = 11$), and SM16e ($n = 4$). Within the mid- to upper-marsh environment, these communities exhibit a clear elevational gradient: SM13b is found at 4.13 ± 0.19 m asl; SM16a is found at 4.39 ± 0.10 m asl; SM16c is found at 4.52 ± 0.21 m asl; SM16e is found at 4.71 ± 0.17 m asl; and SM28 is found at 5.34 m asl. The previous survey of Cambusmore Lodge identified the same five communities.

### 3.2 Classifying vegetation communities using NDVI values

NDVI values extracted using the three different methods (high-resolution and low-resolution imagery of surveyed quadrats and using the existing vegetation polygons) show broadly similar patterns (Table 4, Fig. 5). For the most part, saltmarsh communities have mean NDVI values greater than 0.5. The main exceptions to these are NDVI values derived from the polygons, which typically plot lower. For example, SM10 at Skelbo (0.357 ± 0.236), SM13b at both Cambusmore Lodge and Creag Bheag (0.389 ± 0.278 and 0.381 ± 0.270, respectively), and SM19 at Skelbo (0.355 ± 0.278). In the case of SM13b, NDVI values from imagery of our quadrats all give values greater than 0.5. When using only the data collected by combining our UAV surveys with our field surveys, NDVI appears to be able to differentiate saltmarsh communities from mudflat cover. Mudflat communities mostly give markedly lower NDVI values than adjacent vegetated land. The extraction of mudflat NDVI from simulated quadrats, designed to include a variety of mudflat signatures (such as algal communities at Skelbo and fringing vegetation at Creag Bheag) shows expected variability, with some quadrats at Skelbo showing mean NDVI values as high as adjacent saltmarsh communities. However, NDVI values extracted from polygons covering the whole area clearly show that the mean NDVI values are lower (0.159 ± 0.216, 0.029 ± 0.069, and -0.016 ± 0.034 at Skelbo, Creag Bheag, and Cambusmore Lodge, respectively) than saltmarsh vegetation. When combining the data from all three sites, a value of 0.057 ± 0.132 characterises mudflat surfaces.

**Table 3. Summary elevation and NDVI data quadrats in our new survey and extracted from vegetation polygons from the existing 2011 survey (Haynes 2016).**

| NVC Community | 2022 Survey | | | Haynes (2016) polygons | |
| --- | --- | --- | --- | --- | --- |
| | Elevation (m asl) | NDVI (High-Resolution) | NDVI (Low-Resolution) | Elevation (m asl) | NDVI (Polygon extraction) |
| **Skelbo** | | | | | |
| Mudflat | | | | 0.92 ± 0.51 | 0.159 ± 0.216 |
| SM8 | 3.08 | 0.658 ± 0.140 | 0.635 ± 0.091 | | |
| SM10 | | | | 2.45 ± 0.17 | 0.357 ± 0.236 |
| SM13a | 2.89 ± 0.31 | 0.742 ± 0.104 | 0.686 ± 0.097 | 3.05 ± 0.49 | 0.656 ± 0.207 |
| SM13d | 3.13 ± 0.23 | 0.801 ± 0.075 | 0.736 ± 0.118 | 2.96 ± 0.21 | 0.633 ± 0.178 |
| SM16b | | | | 3.44 ± 0.28 | 0.779 ± 0.136 |
| SM16c | 3.47 ± 0.16 | 0.825 ± 0.067 | 0.798 ± 0.060 | | |
| SM16e | | | | 4.28 ± 0.63 | 0.812 ± 0.103 |
| SM18a | | | | 3.69 ± 0.23 | 0.774 ± 0.195 |
| SM19 | | | | 2.54 ± 0.84 | 0.355 ± 0.278 |
| SM28 | | | | 4.06 ± 0.29 | 0.718 ± 0.086 |
| **Creag Bheag** | | | | | |
| Mudflat | | | | 2.29 ± 4.10 | 0.029 ± 0.069 |
| SM13a | 3.68 ± 0.82 | 0.698 ± 0.135 | 0.591 ± 0.147 | | |
| SM13b | | | | 3.33 ± 1.08 | 0.389 ± 0.278 |
| SM16b | | | | 5.60 ± 0.26 | 0.698 ± 0.150 |
| SM16c | 2.68 ± 0.62 | 0.772 ± 0.088 | 0.778 ± 0.079 | 4.76 ± 1.09 | 0.671 ± 0.157 |
| SM28 | | | | 6.33 ± 0.75 | 0.650 ± 0.143 |
| **Cambusmore Lodge** | | | | | |
| Mudflat | | | | 2.66 ± 0.46 | -0.016 ± 0.034 |
| SM13b | 4.13 ± 0.19 | 0.803 ± 0.093 | 0.783 ± 0.048 | 3.44 ± 0.22 | 0.381 ± 0.270 |
| SM16a | 4.39 ± 0.10 | 0.770 ± 0.095 | 0.793 ± 0.058 | 4.30 ± 0.39 | 0.741 ± 0.187 |

| SM16c | 4.52 ± 0.21 | 0.790 ± 0.092 | 0.756 ± 0.172 | 4.63 ± 0.20 | 0.745 ± 0.086 |
| SM16e | 4.71 ± 0.17 | 0.770 ± 0.095 | 0.752 ± 0.065 | 5.11 ± 1.16 | 0.603 ± 0.144 |
| SM28 | 5.34 | 0.672 ± 0.137 | 0.599 ± 0.076 | 4.87 ± 0.25 | 0.549 ± 0.144 |


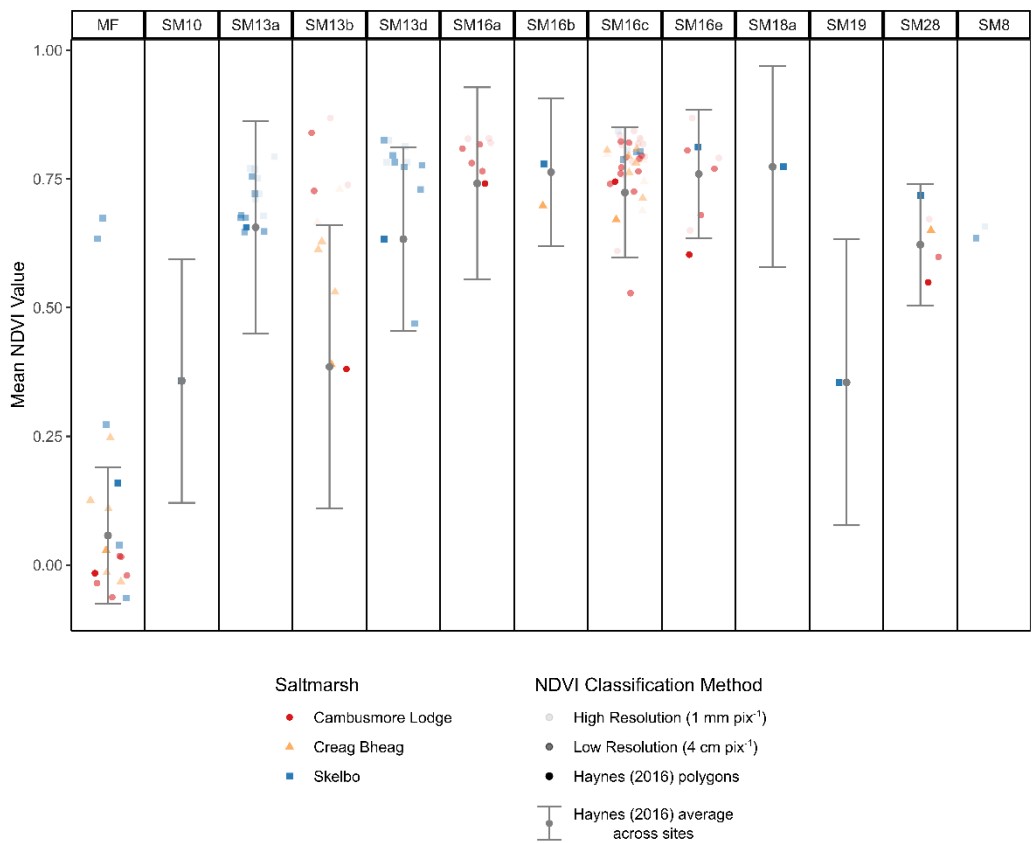

**Figure 5: NDVI values extracted from our quadrats and extracted from polygons from the existing 2011 survey (Haynes, 2016). Where in the 2011 survey the same saltmarsh community is present at multiple sites, these data have been combined and the mean and 1SD range is shown with the grey point and error bar. MF = Mudflat**


### 3.3 Determining NDVI thresholds for area estimates

Based on the NDVI extractions from vegetation, ranges of thresholds were created that would enable saltmarsh area to be estimated. For each site, and each method (high- and low-resolution imagery and existing polygons), ranges were generated using the highest and lowest NDVI values for saltmarsh vegetation communities. The means, 1 standard deviation, and 2
standard deviation ranges between the highest and lowest values were then used as ranges (Table 4).

**Table 4: NDVI ranges used to estimate saltmarsh areas. Note that NDVI has a maximum possible value of 1, and maximum possible values have therefore been constrained to 1.**

| Range | High-Resolution (1 mm pix$^{-1}$) | Low-Resolution (4 cm pix$^{-1}$) | Haynes (2016) polygons |
|---|---|---|---|
| Skelbo | | | |
| Mean | 0.658 – 0.825 | 0.635 – 0.798 | 0.355 – 0.812 |
| 1 Standard Deviation | 0.518 – 0.892 | 0.534 – 0.858 | 0.077 – 0.915 |
| 2 Standard Deviation | 0.378 – 0.959 | 0.453 – 0.918 | -0.115 – 1 |
| Creag Bheag | | | |
| Mean | 0.698 – 0.772 | 0.591 – 0.778 | 0.389 – 0.698 |
| 1 Standard Deviation | 0.563 – 0.860 | 0.444 – 0.857 | 0.111 – 0.848 |
| 2 Standard Deviation | 0.428 – 0.948 | 0.297 – 0.937 | -0.167 – 0.998 |
| Cambusmore Lodge | | | |
| Mean | 0.672 – 0.826 | 0.599 – 0.756 | 0.381 – 0.714 |
| 1 Standard Deviation | 0.535 – 0.916 | 0.523 – 0.928 | 0.111 – 0.928 |
| 2 Standard Deviation | 0.398 – 1 | 0.447 – 1 | -0.159 – 1 |

## 3.4 Estimates of saltmarsh extent

### 3.4.1 Previously mapped saltmarsh extent

 Due to UAV restrictions, full saltmarsh areas were difficult to map at the sites. To enable comparisons between established data and newly generated saltmarsh maps, the previously mapped vegetation distribution maps were cropped to the areas covered by our UAV surveys. The previously mapped areas that are comparable with our mapped areas are 27248.13 m$^2$ at Skelbo, 6818.48 m$^2$ at Creag Bheag, and 44046.11 m$^2$ at Cambusmore Lodge.

### 3.4.2 Saltmarsh extent calculated using NDVI classifications

Saltmarsh extent estimates calculated from a combination of NDVI classifications are shown in Table 5. The lowest extent estimates inevitably come from methods using the lowest ranges of values (the means of the highest and lowest NDVI values for saltmarsh vegetation communities, using either the high- or low-resolution datasets but never the Haynes (2016) polygons) where they are also constrained by the smallest tidal range (the Wick tidal data). Using these values, the lowest saltmarsh extent estimates are 8795.91 m$^2$ at Skelbo and 2077.52 m$^2$ at Creag Bheag. At Cambusmore Lodge, the Wick tidal data does 355 not reach the saltmarsh; the lowest estimate, using the Aberdeen tidal data, is therefore 9585.98 m$^2$.

The highest values are those using the widest range of values (the 2 standard deviation range extracted from the Haynes (2016) polygons) without any independent tidal constraints. In most cases, these estimates are extreme over-estimations which encompass the entire mapped area; they fail to differentiate between saltmarsh vegetation communities and the surrounding mudflats and woodland vegetation. For example, at Cambusmore Lodge the 2 standard deviation ranges under both tidal 360 constraints (without a lower limit) contain such a wide range of NDVI values that the area estimates include the whole mudflat region.

Assuming that the established mapped areas are reliable estimates of saltmarsh extent, they can be used to evaluate the success of our new UAV-derived methods to approximate saltmarsh extent. Using this criterion, at all sites the closest estimates are derived from estimates made using NDVI ranges spanning 1 standard deviation. At all three sites the 1 standard deviation 365 ranges both over- and under-estimate saltmarsh extent, with variation depending on the tidal constraints applied. In percentage terms, six estimates fall within 10% of the existing estimates. Of these, five are derived from 1 standard deviation estimates (Fig. 6).

**Table 5: Estimations of saltmarsh vegetation area (m$^2$) combining different methods of NDVI classification and using different tidal constraints.**

| | | High-Resolution (1 mm pix$^{-1}$) | Low-Resolution (4 cm pix$^{-1}$) | Haynes (2016) polygons |
|---|---|---|---|---|
| **Skelbo** | | | | |
| **Aberdeen tidal gauge** | **Mean** | 13850.75 | 12110.44 | 18757.36 |
| | **1SD** | 25763.13 | 21326.74 | 33965.67 |
| | **2SD** | 31435.45 | 30075.94 | 35215.62 |
| **Wick tidal gauge** | **Mean** | 10109.81 | 8795.91 | 11033.41 |
| | **1SD** | 17565.08 | 15294.54 | 21823.49 |
| | **2SD** | 19404.89 | 18791.99 | 32066.44 |
| **No tidal constraints** | **Mean** | 14938.17 | 13080.95 | 20442.93 |
| | **1SD** | 29555.43 | 23215.14 | 44664.75 |

| | | | | |
|---|---|---|---|---|
| | **2SD** | 48953.28 | 42684.01 | 53063.03 |

**Creag Bheag**

| | | | | |
|---|---|---|---|---|
| Aberdeen tidal gauge | **Mean** | 2589.13 | 5737.94 | 5605.94 |
| | **1SD** | 8246.22 | 9871.32 | 12329.10 |
| | **2SD** | 10175.62 | 11003.09 | 27185.78 |
| Wick tidal gauge | **Mean** | 2077.52 | 4603.69 | 4594.31 |
| | **1SD** | 6674.62 | 8046.09 | 10019.4 |
| | **2SD** | 8306.19 | 9000.00 | 22762.62 |
| No tidal constraints | **Mean** | 7059.15 | 17640.66 | 17831.91 |
| | **1SD** | 23636.99 | 28329.11 | 32363.39 |
| | **2SD** | 29271.31 | 31190.65 | 39540.87 |

**Cambusmore Lodge**

| | | | | |
|---|---|---|---|---|
| Aberdeen tidal gauge | **Mean** | 15356.46 | 9585.98 | 13462.87 |
| | **1SD** | 26393.26 | 26728.64 | 31845.21 |
| | **2SD** | 29101.51 | 28393.86 | 100314.99 |
| Wick tidal gauge | **Mean** | NA | NA | NA |
| | **1SD** | NA | NA | NA |
| | **2SD** | NA | NA | NA |
| No tidal constraints | **Mean** | 24523.96 | 16444.65 | 28618.73 |
| | **1SD** | 43434.25 | 44185.02 | 58994.79 |
| | **2SD** | 52038.93 | 49229.57 | 131612.9 |


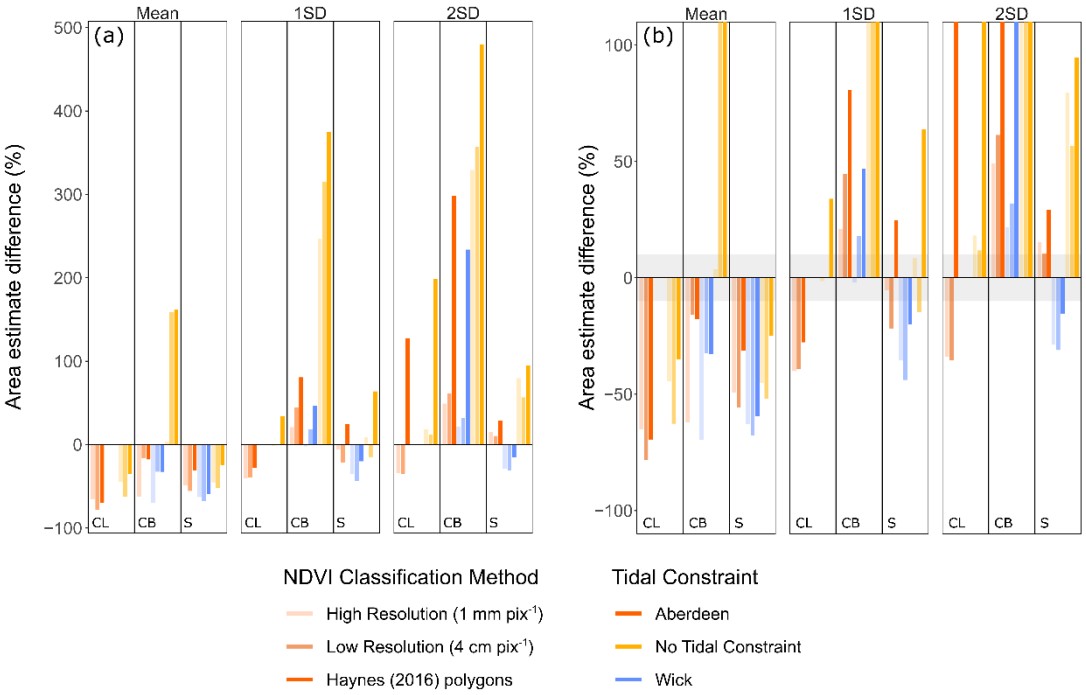

**Figure 6:** **Relative area estimates between the existing Loch Fleet saltmarsh maps (Haynes, 2016) and our new UAV NDVI-derived estimates. Differences are presented as the percentage difference between the two estimates, where 0 marks the existing estimates and positive values denote methods where new estimates are greater than the existing maps. (a) The full range of relative estimates; (b) Estimates cropped to within ± 100%, the grey box represents ± 10%. Codes refer to Cambusmore Lodge (CL), Creag Bheag (CB), and S (Skelbo).**

### 3.5 Estimates of OC storage in saltmarshes

Estimates of OC stocks in Loch Fleet saltmarshes vary significantly with the different approaches to estimating saltmarsh extent (Fig. 7). Because the Wick tidal gauge data generates the smallest area extents (particularly evident at Cambusmore Lodge, where the Wick HAT limit fails to reach the saltmarsh environments at all), those areas generate the smallest OC stock estimates: 181.2 ± 39.3 tonnes OC at Skelbo (using the mean NDVI values from the low-resolution flights) and 42.8 ± 9.3 tonnes OC at Creag Bheag (using the mean NDVI values from the high-resolution flights). Conversely, the highest OC stock estimates are generated when applying no tidal constraints; OC stock estimates are around an order of magnitude higher using these area estimates. At Skelbo the maximum estimate is 1,092.9 ± 236.7 tonnes OC, at Creag Bheag the estimate is 814.4 ± 176.7 tonnes OC, and at Cambusmore Lodge the maximum estimate is 2,710.7 ± 588.1 tonnes OC. Using these estimates, the total quantity of OC stored in the three saltmarshes in Loch Fleet ranges between a minimum of 251.0 ± 46.1 tonnes OC, and a maximum of 3,995.9 ± 620.7 tonnes OC.

Across the three sites, it appears that the precision of NDVI estimation methods is an important variable when quantifying OC storage. At all sites, the differences in mean values between the high- and low-resolution imagery is minor when compared

with the estimates from the broader, less precise Haynes (2016) polygon estimates (Fig. 7). Not only do the polygon estimates have a significantly higher range than the high- and low-resolution imagery, but the means are also uniformly higher.

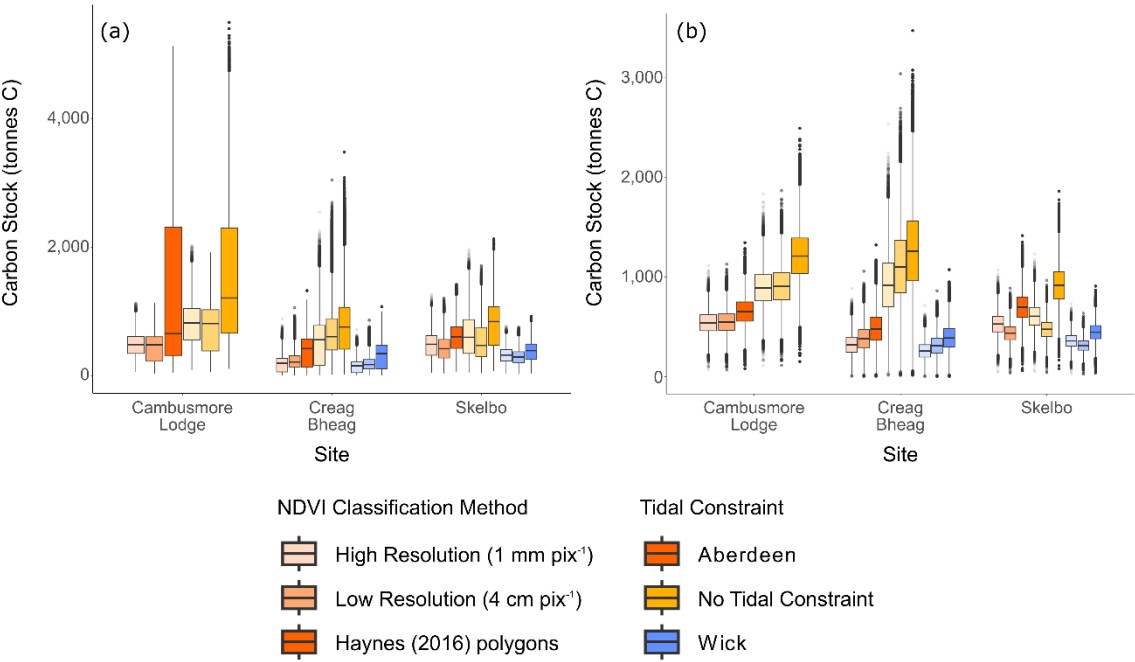

**Figure 7: Estimates of OC stocks generated from the MCMC OC models. (a) Data include all range estimates (Mean, 1SD, and 2SD NDVI ranges); (b) Data for the 1SD range only.**

Assuming that existing area estimates offer an opportunity to assess the accuracy of our new estimates (see Sect 3.3.3), it is likely that the estimates derived from 1 standard deviation NDVI ranges are most reliable. It also appears that our new high- and low-resolution images yield more useful NDVI ranges than those derived from existing vegetation area polygons. Using these NDVI ranges the OC storage estimates encompass a significantly smaller range; the highest possible values are only double the smallest. The most likely estimates come when including the Aberdeen tidal gauge data as a constraining variable.

When using this, the high-resolution imagery provides an estimated total for our three saltmarshes of 1,341.1 ± 228.9 tonnes OC, and the low-resolution imagery provides an estimated value of 1,271.5 ± 257.4 tonnes OC (Table 6). This compares with a stock of 1,608.9 ± 233.8 tonnes OC based on existing saltmarsh vegetation maps (Haynes, 2016), a reduction of approximately 15-20%.


**Table 6: OC stock estimates from our new UAV-derived surveys, showing stock estimates for each saltmarsh using the 1 standard deviation ranges of NDVI values. The total OC stock for the three surveyed saltmarshes are also calculated. Also included are OC**
**stock estimates based on existing saltmarsh estimates for the three saltmarshes (Haynes, 2016).**

| Tidal Constraint | Resolution | OC Stock (kg) | | | |
|---|---|---|---|---|---|
| | | Skelbo | Creag Bheag | Cambusmore Lodge | Total |
| Aberdeen | High-Resolution | 530.6 ± 115.4 | 324.1 ± 110.7 | 486.4 ± 163.7 | 1,341.1 ± 228.9 |
| | Low-Resolution | 439.2 ± 85.5 | 388.0 ± 132.5 | 444.3 ± 203.4 | 1,271.5 ± 257.4 |
| Wick | High-Resolution | 361.8 ± 78.7 | 262.3 ± 89.6 | NA | 624.1 ± 119.3 |
| | Low-Resolution | 315.0 ± 68.5 | 316.25 ± 108.0 | NA | 631.3 ± 127.9 |
| No Tidal Constraint | High-Resolution | 608.7 ± 132.4 | 929.1 ± 317.4 | 823.8 ± 300.9 | 2,361.6 ± 456.9 |
| | Low-Resolution | 478.1 ± 104.0 | 1,113.5 ± 380.4 | 754.2 ± 345.0 | 2,345.8 ± 523.9 |
| | Haynes (2016) | 561.2 ± 122.0 | 140.4 ± 30.5 | 907.2 ± 197.1 | 1,608.9 ± 233.8 |

## 4. Discussion

### 4.1 Constraining saltmarsh extent using high-resolution digital surface models

Saltmarsh communities and sediments are regulated by their position in the tidal range. However, often coastal elevation profiles are restricted by low-resolution elevation estimates. UAV surveys offer a promising opportunity to develop high-
resolution, geospatially accurate elevation estimates rapidly using several different methods.

The elevation profiles generated here suffer from two restrictions. Firstly, they are not direct elevation estimates. Instead, the elevation profiles have been constructed using structure-from-motion methods which use complex geometric calculations, combined with high-precision UAV geopositioning data, to estimate relief and elevation. Nevertheless, when combined with the high accuracy real-time kinematic data they produce highly accurate and repeatable elevation maps. Secondly, the elevation
models generated from the majority of affordable UAVs currently on the market are digital surface models; rather than mapping the ground, they map the highest photographed surface, in most cases the vegetation. While many saltmarsh vegetation communities are comprised of low-lying plants that do not extend far from the surface, others (e.g. the *Pucinellia* and *Festuca* grasses of the mid-marsh zones) can grow to heights which may have a non-negligible effect on the reconstructed

elevation (DiGiacomo et al., 2020). As LIDAR becomes an increasingly affordable attachment to UAVs, the potential for accurate elevation mapping will be greatly enhanced, allowing more precise elevation maps and allowing them to be more accurately related to tidal data.

Additionally, few areas have site specific tidal data and in this study data from the nearest two tidal gauges (Aberdeen and Wick) were used to estimate tidal conditions within Loch Fleet. It is clear that neither of the two datasets are wholly applicable to the whole Loch Fleet shoreline. At Skelbo both the Aberdeen and Wick HAT values overlay onto the mapped saltmarsh communities with success. Conversely, at Creag Bheag neither tidal value fully encompasses the range and extent of saltmarsh communities mapped during the surveys. At Cambusmore Lodge, the Wick tidal data does not encompass any of the mapped saltmarsh, whereas the Aberdeen data appears to correspond well with the saltmarsh vegetation communities but does not cover saltmarsh communities at higher elevations (if compared with the established Haynes (2016) vegetation distributions). It is likely that local geomorphological and oceanographic factors drive these variations. Skelbo is the closest site to the North Sea and the most exposed, meaning that regional tidal patterns are likely represented with little localised distortion. Cambusmore Lodge faces east towards the North Sea, but is the furthest inland, and only the Aberdeen tidal range encompasses the mapped saltmarsh areas. Conversely, neither tidal range at Creag Bheag corresponds with mapped saltmarsh vegetation communities. It is likely that this is due to complex tidal dynamics; Creag Bheag faces west and is situated in a sheltered corner of Loch Fleet, and complex circulation of water around the loch is likely to modulate the dominant tidal patterns. As new, affordable tools for localised monitoring of tidal dynamics (e.g. Mini Buoys; Balke et al., 2021) become available, our ability to link tidal data with remote sensing data will improve significantly and allow us to better map saltmarsh areas.

## 4.2 Constraining saltmarsh extent using NDVI values

The different methods for extracting NDVI values yield different results. Of particular note is that the NDVI estimates extracted from the previously mapped vegetation areas (Haynes, 2016) consistently yield wider ranges of NDVI values than those generated from our quadrats. This is explicable for two reasons. Firstly, the polygons represent an extrapolated vegetation zone based on systematic infield surveys, and in reality are likely to incorporate a range of vegetation communities. When using those mapped units to extract NDVI values, it is inevitable that they will produce a wider distribution of values compared with NDVI values extracted from quadrat areas with a known vegetation community. Secondly, the earlier survey was conducted in 2011. Saltmarshes are highly dynamic ecosystems (Ladd, 2021), and it is probable that some of the vegetation units have migrated between the initial survey and the new UAV mapping. Nevertheless, there remains clear synergies between the existing surveys and the new data. However, given the wider range of NDVI values it is inevitable that this method produces less reliable saltmarsh extent estimates than our other methods.

NDVI values generated from the high- and low- resolution imagery of our surveyed quadrat areas are broadly comparable with one another, with only slight differences. Many of the differences can be attributed to slight differences in the area being analysed; while the high resolution images of the quadrat accurately represent the vegetation community being surveyed, the low-resolution analysis relied upon accurate GPS positioning to estimate the location of the quadrat. The high-resolution

imagery of the quadrat areas proved a time-consuming exercise, and several quadrats lacked the sufficient unique tie points for photogrammetric methods to successfully stitch together images. The lack of significant differences with the low-resolution imagery suggests that such high-resolution work is unnecessary to accurately estimate saltmarsh extent.

It is clear that mudflat environments can, on average, be differentiated from saltmarsh environments. Where bare mudflat environments exist, such as at Cambusmore Lodge, those mudflats have markedly lower NDVI values than nearby vegetated area (Fig. 5). At other sites, where the transitionary zone is less sharply delineated, the NDVI separation is less successful. At Skelbo, objects on the mudflat that are evidently not saltmarsh communities (based on in-field examination and the RGB images) nevertheless yield NDVI values indistinguishable from saltmarsh vegetation. Similarly, at Creag Bheag nearshore

algal communities on the mudflat produce NDVI values similar to saltmarsh communities, but in areas where pioneer saltmarsh communities also appear to be forming (Fig. 3). At each of the three sites studied here, a sharp elevational change demarcates mudflats from the major saltmarsh units, and implementing a minimum elevation threshold is likely to provide enhanced reliability to saltmarsh area estimates.

Using NDVI values alone to constrain saltmarsh extent shows significant limitations. At all three sites, when using the largest

range of NDVI values (the 2 standard deviation range) the saltmarsh extent began to encompass both the fringing terrestrial vegetation (most obviously trees clearly outside the tidal range) and the unvegetated mudflat. It is evident from visual examination of the NDVI orthomosaics that the canopies of trees have similar NDVI signatures to the saltmarsh communities (Fig. 3), and without using tidal data to constrain the saltmarsh extent estimates it is impossible to delineate saltmarsh from fully terrestrial environments with tree cover. Similarly, the 2 standard deviation range incorporates low NDVI values that are

characteristic of bare mudflat surfaces, making it impossible to delineate saltmarsh communities using NDVI values alone.

Under the assumption that previous mapping exercises present accurate estimates of saltmarsh extent, it is interesting that at each of the three sites a different UAV estimation method yields the closest estimate. At Skelbo, the 1 standard deviation range generated from the high-resolution quadrat imagery produces a saltmarsh estimate that is within 94.6% of the previously mapped area. At Cambusmore Lodge, on the other hand, the closest estimate (100.3% of the mapped area) and second closest

(98.6%) are derived from the 1 standard deviation range of the low- and high-resolution data, respectively, but without any tidal constraint. Interestingly, comparing the previous maps with newly generated ones the lack of tidal constraint clearly incorporates fringing trees while excluding a large area previously mapped as SM16e and SM28 communities. At Creag Bheag, some estimates are close to the previous extent estimation. However, comparisons of the maps show that without a tidal constraint the NDVI data alone are not successfully separating saltmarsh vegetation from the surrounding woodland, and

incorporating tidal constraints yields highly inaccurate maps with very low saltmarsh estimates. It is likely that this is due to abnormal tidal data at this site which, due to local loch geometry, cannot be accurately modelled using data from the nearest tidal gauges (see Sect. 4.1).

It therefore seems apparent that using NDVI data combined with contemporary vegetation surveys produces quantifiable data that can be used to map saltmarsh habitats. The 1 standard deviation ranges generated here appear to produce applicable

constraining NDVI values that, in combination with appropriate constraints (e.g. appropriate tidal data and high-resolution

DSMs) allow saltmarsh environments to be well-constrained. The success of the 1 standard deviation ranges, in comparison with the broader 2 standard deviation limits, is likely due to the removal of irrelevant NDVI signatures within the imagery; for example, if images include small portions of bare ground exposed by the movement of the vegetation, they will incorporate a wider range of NDVI values. This allows the NDVI data to successfully demarcate not only the fringing mudflat communities,

but it also allows maps to identify and distinguish unvegetated creeks and pans which may be difficult to accurately map and account for using field methods.

## 4.3 Estimating saltmarsh OC storage

Saltmarshes represent natural sinks of OC and are becoming increasingly recognised as valuable natural areas that can potentially mitigate climate change (Duarte et al., 2013; Burden et al., 2019). Many saltmarsh systems are small and isolated,

and identifying and fully investigating all sites for inclusion into OC stock assessments is a costly and time-consuming endeavour. Nevertheless, to accurately account for OC storage at the national scale new approaches are required to rapidly map and monitor these dynamic environments. It is apparent that NDVI values generated from high- and low-resolution UAV imagery generate broadly comparable estimates of OC storage, whereas the NDVI values generated using mapped vegetation data typically generate higher estimates with more variability (*Sect. 4.2.1*). The UAV-derived estimates of saltmarsh vegetation

extent, and by extension modelled estimates of OC storage in saltmarsh soils, have generated a wide range of OC stocks for the saltmarshes within Loch Fleet. However, when constraining the data to give the best possible estimates of saltmarsh area, the total estimated OC storage in the three marshes surveyed here is approximately 20% lower than estimates generated using area estimates from previous survey data. The reduction in saltmarsh extent and OC storage estimation is most dramatic at Cambusmore Lodge, where our new UAV estimates are approximately half of those in the previous survey (Haynes, 2016).

One explanation for this is that the previous survey identified a large, contiguous SM16e community beyond the drift-line which we did not identify in our field survey, and the area of which has markedly lower NDVI values that fall outside the ranges identified in our field survey. Even in those estimates that do not include tidal constraints, the SM16e communities are identified only sporadically, and the NDVI values do not indicate that contiguous communities exist in this area. It is plausible that this reflects a true change in the distribution of saltmarsh vegetation in the decade since the initial survey was conducted.

Alternatively, it may suggest that the vegetation at the time of survey was under stress and therefore presenting lower NDVI values. We elected to sample in June as NDVI signals were expected to be strongest (Sun et al., 2018). However, plant stress caused by environmental variables at higher elevations (for example, salinity increases if not inundated frequently and increased evaporation during warm summers) may mediate NDVI signals (Doughty and Cavanaugh, 2019). It is therefore possible that saltmarsh communities continue to exist at higher elevations but are exhibiting lower NDVI values than expected.

A secondary reason for the marked difference in saltmarsh estimation between the two surveys is that the UAV imagery is successfully identifying creeks and pans. These are areas where organic-rich saltmarsh soils, and the OC that they contained, have been eroded and the OC lost. These areas are difficult to map using traditional surveying methods. The high-resolution UAV data proves adept at identifying these features and removing them from the OC stock calculations.

The difference in the saltmarsh areal extent between the previous survey (Haynes, 2016) and the data presented here potentially has significant consequences for our understanding of saltmarsh OC storage. In the three saltmarshes of Loch Fleet the OC stock estimates calculated using the NDVI approach are 15 – 20% lower than that using the previous surveyed extents (Haynes, 2016). The difference suggests that either the saltmarshes have reduced in size since the 2011 survey, or the original survey overestimated the extent of saltmarsh habitat. The decrease in size in the 11 years since the original survey would equal a loss 1.4 – 1.8% per year, far exceeding the global average of 0.28% per year (Campbell et al., 2022). Further, the differences in areas are not restricted to the seaward edge of the marshes, and therefore erosion cannot be the sole driver of this change. Consequently, it is likely that the Haynes (2016) overestimated the extent of saltmarsh habitat in Loch Fleet that will have led to an overestimation of the OC stocks of individual saltmarshes.

The current methodologies used to estimate saltmarsh OC stocks in Scotland (Smeaton et al., 2022; Miller et al., 2023) place a ± 5% error on the area value derived from the Haynes (2016) survey to account for potential erosion or accretion. This approach, in combination with the MCMC framework (as used in this study), results in Scotland's saltmarshes storing 1.15 ± 0.21 Mt of OC, with the 5$^{th}$ and 95$^{th}$ percentile values ranging between 0.71 and 1.40 Mt OC (Miller et al., 2023). If the 15 – 20% error observed in the Loch Fleet saltmarshes is systemic across all Scottish systems, the OC stock would be reduced by between 0.17 and 0.23 Mt, which is still within the calculated uncertainties presented in Miller et al., (2023). Nevertheless, to assure accurate OC stock estimates, in future consideration should be taken to apply an appropriate error value when utilising the Haynes (2016) data.

## 4.4 Applications to management and monitoring of blue carbon environments

Improving our ability to remotely map and understand blue carbon environments is essential for understanding the distribution and storage of OC in coastal areas (Malerba et al., 2023). Due to their potential to act as nature-based solutions, it is important to consider whether this approach could be used to monitor natural blue carbon systems and assess how interventions (e.g. management actions and realignment projects) are affecting the environments, alongside the impacts of natural pressures such as changing sea level.

We believe that the speed with which these surveys can be undertaken, alongside the technology which permits identical flight patterns and the low processing cost of this method, produces an excellent tool for the long-term monitoring of these vulnerable coastal wetland ecosystems and environments. With such high-resolution data now available (4 cm pix$^{-1}$ in this study), there is scope to monitor at high spatial and temporal resolution the dynamism of saltmarsh environments. In particular, using a combination of NDVI and DSM data provides a new potential to monitor the both the horizontal accretion and erosion of saltmarsh features. Monitoring accretion in this way could provide a new method for assessing the success of management interventions and provide information on the rates of change they are effecting. Monitoring patches of erosion (such as pans and creeks) could also provide important information on dynamic processes that may cause the loss of saltmarsh OC. The flexibility and ease of use of UAVs would allow for rapid assessments after discrete erosion events, for example after winter storms (Leonardi et al., 2015). Extending this method to the adjacent mudflat communities, the high-resolution DSMs produced

through the regular application of this method could provide important information on bed level dynamics which are essential for identifying areas where saltmarsh vegetation might colonise in the future (Willemsen et al., 2022).

Looking towards other blue carbon systems, one of the main principles of our technique is the incorporation of high-resolution DSMs with local sea level data to constrain the intertidal zone. This type of distinct elevational zone is not something that is applicable to many other ecosystem types but provides a clear demarcation in coastal communities and blue carbon habitats. Mangroves are one of the most globally significant blue carbon ecosystems, with large pools of above-ground and below-ground OC storage (Rovai et al., 2018). Like saltmarsh systems, mangroves are intertidal and regularly inundated by seawater. Existing work using hyperspectral signatures, combined with LiDAR data, suggests that existing approaches are able to distinguish between different mangrove species, often to tree level (Yin & Wang, Cao et al., 2021). Given this, we consider it likely that with some adaptations our approach could successfully delineate mangroves. Due to the height of the mangrove shrubs and trees, LiDAR capabilities would be required to distinguish the sediment surface height and its position in the tidal range, and studies would need appropriate tidal data in order to constrain elevation data which may not always be locally available to the extent that it is in the UK. The accessibility of mangrove environments in comparison with saltmarshes, and the spatial scales on which they are distribute, may also reduce the relative utility of UAV technology in comparison to satellite data. Nevertheless, we consider it reasonable that small mangrove systems could be successfully and accurately mapped using the methods outlined here.

## 5. Conclusions

Saltmarsh environments are natural sinks of OC located at the transition between the terrestrial and marine systems. Their function as OC sinks and long-term OC stores provides natural reservoirs that can act to potentially modulate atmospheric carbon dioxide ($CO_2$). To fully understand their current role in modulating atmospheric $CO_2$, and their future roles in sequestering and storing atmospheric emissions, accurate and high-resolution mapping of saltmarsh cover is required. Given that saltmarsh environments may be identified by distinctive and characteristic vegetation communities, there is therefore scope for using new surveying tools for the identification and differentiation of vegetation to remotely identify saltmarsh environments without the need for detailed infield surveys.

This study has investigated the use of high-resolution multispectral data derived from uncrewed aerial vehicles to identify and map saltmarsh communities around Loch Fleet, northern Scotland. Using a combination of field vegetation surveys and high-resolution (1 mm pix$^{-1}$ and 4 cm pix$^{-1}$) UAV imagery, the NDVI vegetation index has been assessed for its use in demarcating saltmarsh communities from adjacent marine and fully terrestrial environments. The differences between estimates from high- and low-resolution imagery are minor, and therefore the time-intensive process of taking and processing high-resolution imagery of individual quadrats is not necessary. Accurate mapping of saltmarshes has a significant impact on estimations of soil OC storage, and over-estimations of saltmarsh extent based on low-resolution satellite data and field surveys risk significantly over-estimating the magnitude of saltmarsh OC stores. Future work calibrating and ground truthing UAV signals

may help to map saltmarsh zonation patterns, which can have marked impacts on soil OC density (Austin *et al.*, 2021).
Developments in image classification techniques, such as machine learning approaches, may allow us to map individual saltmarsh communities and further refine our understanding of the relationships between above ground biomass and below ground OC storage.

UAV-based methods therefore provide a rapid approach to mapping and monitoring change in saltmarsh environments. Saltmarshes are dynamic environments and, given their importance for OC storage, repeated surveys using consistent methods
are essential for fully understanding the processes and changes that affect them. A standardised approach is therefore required to enable comparisons and reliable change detection, and the methodology presented here can be readily implemented with little cost. As UAV technology advances (for example, developments in  hyperspectral imaging capabilities), and we are better able to monitor local tidal conditions, our ability to combine aerial imagery with accurate tidal data will further improve our ability to accurately map saltmarsh environments and estimate the OC their soils contain. Future work calibrating and ground-
truthing UAV signals with a variety of saltmarsh characteristics, including carbon sequestration and storage, would additionally enhance our ability to map and understand saltmarsh development.

## Data Availability

All data are available on request from the corresponding author.

## Author Contributions

WH and CS led the conception and developed the project. WH, CS, LM undertook the fieldwork; WH undertook the data processing. WH wrote the first draft of the manuscript with assistance of CS. WA secured the funding for the project. All authors contributed to the manuscript revision and approved the submitted version.

## Competing Interests

The authors declare that they have no conflict of interest.

## Acknowledgements

This work was jointly supported by funding from the Highland Council through the Climate Action Coastlines Project (University of St Andrews grant reference no. SGS0-XGLH48) and by the Natural Environment Research Council (grant NE/R010846/1) Carbon Storage in Intertidal Environments (C-SIDE) project. We would like to the thank NatureScot for
assisting in gaining access to the sites and facilitating the use of the UAV.

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
