# Peer review of "UAV approaches for improved mapping of vegetation cover and estimation of carbon storage of small saltmarshes: examples from Loch Fleet, northeast Scotland"

_EGUsphere, 2023_

## Author Response (AR1)

| Reviewer Comment | Response | Lines of Change (track change view off) | Lines of Change (track change view on) |
|---|---|---|---|
| The paper describes an approach combing vegetation survey, UAV data and tidal data to estimate saltmarsh extent and saltmarsh organic carbon storage. It is certainly intended to be a method development study but has significant amount of work on the effects of areal estimates on organic carbon storage estimates. Do the authors use the estimate of OC storage as a way to assess the reliability of different approaches in extracting saltmarsh extent? As a result, I am less certain of the paper's objective(s). In any case, I see some values of this work but would like to see some improvements. | We have chosen to use carbon storage as a proxy to show the impact of our proposed method, as blue carbon storage is a major question in studies of near-coastal environments and because it fits within the scope of the special issue. We do not intend to use it as a means of testing the reliability of our approaches, but rather to demonstrate the impact of different areal extent estimates on estimates of carbon storage, which has knock-on effects on issues such as nature-based solution approaches and carbon budgeting. We have adjusted our third objective to better reflect this approach. | 77-78 | 83-84 |
| I found the five parts in Results section confusing and not appropriate to respond to the three objectives the authors raised in Introduction section. Especially for section 3.1, the authors should explain why it was included in Results section. Section 3.2 and 3.3 should be moved to Methods section, as it describe how to classify vegetation communities and estimate areas. | We appreciate that there is a case to be made for including Sections 3.2 and 3.3 in the methods section rather than the results section. However, we believe that each of these sections (3.1 to 3.3) represent the primary results of our field and UAV study and, although they do form the basis for the later mapping results (essentially forming our training dataset), they are best placed in the Results section rather than the Methods section. | | |
| How do the authors define the saltmarsh environments? Does it mean saltmarsh ecosystems? | We appreciate the reviewer for highlighting to us our interchangeable use of environments, vegetation, ecosystems, etc. We have worked through the manuscript and standardised usage so that "saltmarsh vegetation (communities)" refers only to vegetation, whereas "saltmarsh environments" refers to the | Throughout | Throughout |

| | | | |
|---|---|---|---|
| | broader coastal system that includes vegetation characteristics, tidal characteristics, and the underlying saltmarsh sediments. | | |
| There is a fundamental issue here that the paper did not explore. When classifying saltmarsh vegetation, the authors used the National Vegetation Classification (NVC) scheme. It seems not require training data and validation data as well, so how the mapping accuracy can be achieved? This relates to the following estimation of saltmarsh extent. | The NVC system is an established and extensively used system based on an extensive database of vegetation data collected across the United Kingdom, representing almost all communities present in the UK. This classification scheme is based on approximately 35,000 samples, which removes the need for individual studies to develop training datasets. The MAVIS system then uses multivariate methods to statistically assign a newly surveyed vegetation community to an established existing community. We've restructured the sentence in section 2.5 explaining this to make it clearer. | 157-159 | 176-178 |
| In Introduction section, the authors point out the location (elevation) of saltmarshes, that is, saltmarshes form between the high astronomical tide (HAT) and mean tidal level (MTL). In section 2.5.1, the authors stated that the saltmarsh area was estimated by calculating the area that inundated under each tidal condition. In this context, the authors focused on three tidal contexts: High Astronomical Tide (HAT), Mean Low Water Springs (MLWS), Mean High Water Neaps (MHWN). What is the connection between MTL, MLWS, and MHWN? Why the authors recognize the saltmarshes are expected to always be inundated? | We thank the reviewer for highlighting that we have not fully explained the rationale behind these choices. These metrics do not mean that saltmarshes will always be inundated, but they reflect different periodicities of inundation which are thought to exert control on saltmarsh vegetation formation. We take the HAT because this represents the highest elevation that might be expected to be periodically (but not necessarily annually) inundated by sea water, and we use this to constrain the upper limits of the possible saltmarsh environments. We could not use MTL, as that data isn't available from the UK Tidal Gauge Network from which we obtained the tidal data, and this isn't necessarily the most useful value, and we have modified the text to more explicitly focus on | 49-53; 194-199; Table 2; Figure 5 | 49-53; 214-219; Table 2; Figure 5 |

| | the HAT, which we ultimately used for the analyses. We have changed several sections to account for this: the section in the Introduction introducing tidal ranges as key threshold for saltmarsh formation; Table 1 and Figure 4 (we ultimately don't use MHWS in our later analyses, so we have removed the data); and we have restructured Section 2.5.1 to more explicitly explain our use of tidal data. | | |
|---|---|---|---|
| Some paragraphs in Section 2.5 and Section 2.6 should be connected. | We have reworked sections 2.6 and 2.7 to better explain our methods. Information on calculating our NDVI ranges has moved from Section 2.7 into Section 2.6.2, and Section 2.7 now only describes how we combine the data from 2.6.1 (tidal data) and 2.6.2 (NDVI data) to get our final estimates of saltmarsh area – these sections should now be better/more satisfactorily linked together. | 215-219; 231-239; 248-254 | 242-246; 258-266; 276-282 |
| How to classify the saltmarsh vegetation? By using NVC scheme or NDVI extraction values described in section 2.5.2? The authors should add more details about vegetation classification. | We have added some content to the start of section 2.6.2 to explicitly explain the relationship between the vegetation survey data, the subsequent NVC classification, and the use of that data to develop relationships between vegetation community and NDVI signatures. This is later expanded upon in Section 3.2. | 214-219 | 242-246 |
| For the first objective of this work, the authors stated to delineate saltmarsh habitats. Vegetation composition is a key component of saltmarsh habitats, however, the related content is not well depicted in this paper. Please change the objective 1 more precisely. | Objective 1 has been refined in response to comments from another reviewer, and in response to the terminology in point one of this review. | 74-78 | 78-81 |
| Do the authors used the same OC storage estimation method with Haynes et al., 2016? Section 2.7 need more related descriptions. | Haynes (2016) only mapped saltmarsh vegetation but did not estimate carbon storage; to create a comparison point for our carbon storage case study, we have therefore simulated the | 278-280 | 323-325 |

| | carbon storage based on their area estimates, using an identical method to the one we used to simulate carbon storage for our new area estimates. We have added a small paragraph at the end of section 2.8 explicitly stating that we used the same method to estimate the carbon storage that would be implied by the Haynes (2016) vegetation maps. | | |
|---|---|---|---|
| Line 35: simple or straightforward? | Amended to straightforward. | 35 | 35 |
| Line 61: saltmarshes are not always small features. | We have amended this sentence to clarify we're referring to UAV's ability to map small saltmarsh areas. | 65 | 68 |
| Line 94-98: these three sentences are not suitable for the study area section. | We have moved these lines into Section 2.1, which is a reworked description of the broader area. We now have an additional section (2.2) with more explicit and detailed site summaries. | 94-99 | 101-105 |
| Section 2.3: this section only describes how to design and collect UAV data. Seems better to change to UAV data collection. | We have amended this as recommended. | | |
| Line 136-137: please state what parameter did the internal sunlight sensor process. | We have amended this sentence to include the solar irradiance values collected by the UAV. | 144-146 | 162-164 |
| Table 1: the tidal ranges are the mean values? | We have added some additional context at the first point where we discuss the tidal data (section 2.6.1). | 195-198 | 214-217 |
| Line 191: typo, rewrite this sentence. | We cannot identify any typo around line 191. | | |
| Figure 4: the Wick HAT in middle 2 figures is not identical to the legend color scheme. | We have amended Figure 4 in two ways. Firstly, we have removed the other tidal constraints, which have ultimately not been used in further analysis. The middle two panels now show the Aberdeen (left) and Wick (right) HAT marks only. We have also therefore been able to amend the colour scheme, and the bottom two panels are now colour coded to align with the middle panels (red for Aberdeen, blue for Wick). | | |

| | Additionally, we have removed the grey outlines on the polygons in the bottom panels, which were causing colour aberrations where lots of outlines of small patches were showing. This should no longer be an issue.
NOTE: Due to the need to split an earlier Figure, this is now Figure 5 | | |
|---|---|---|---|
| Table 2: keep the same decimal place. | The Elevation data is only produced to the nearest cm, whereas the NDVI data is produced to three decimal placed. Given they are not directly comparable, we don't consider it necessary to amend the elevation to an unknown millimeter precision and we wouldn't like to remove some of the detail in the NDVI data.
NOTE: Due to the addition of another Table, this is now Table 3 | | |
| Figure 5: abbreviation of SM10,…SM8 should be clarified. | We have added a sentence in section 2.5.1, clarifying that they SM-X coding that appears through the paper represents a saltmarsh vegetation community as classified using the NVC. We appreciate the reviewer for highlighting that we had not explicitly stated this. | 161-163 | 181-182 |
| The form of the tables (Table 1-5) are not appropriate. | We'd be happy to amend the tables as requested, however we are not sure what form would be preferable. If the reviewer or editor would like to advise us further, we will make the suggested adjustments. | | |

| Reviewer Comment | Response | Lines of Change (track change view off) | Lines of Change (track change view on) |
|---|---|---|---|
| Having read the manuscript, I am left wondering why the authors did not use a classification method, such as the Random Forest | Thank you for this suggestion, which would be a very interesting approach with our data. In general, our main aim in this | 189-196; 591-593 | 212-219; 640-642 |

| | | | |
|---|---|---|---|
| approach used by Villoslada et al. (2020)? Further justification for not using an approach such this, especially when the authors have the available data, is required. | paper has been to identify and test methods which could be easily and readily implemented for saltmarsh monitoring by a range of stakeholders, either within the research community or government agencies, for example. For this reason, we have opted to try to keep the methodology as simple and as widely applicable as possible, with minimal computing power and with as few variables as possible. We have added a short section in our Methods (Section 2.6) to explain this.

However, this suggestion is incredibly interesting, and we would be very interested in exploring this further in the future with this dataset. It would be very interesting to test whether we could identify our discrete saltmarsh communities in our data using a wider variety of spectral signals and using machine learning approaches, which would represent a significant step forward in our ability to remotely map saltmarsh communities and carbon storage in the UK. We have added a line in the Conclusions pointing towards this potential. | | |
| There is also a need to consider the wider context and implications of the work. For example, it would be interesting to see some discussion of application of the approach in other systems such as mangroves or even restored saltmarshes including managed realignment sites. There have been a number of studies into the use of UAS approaches and blue carbon in these settings, it would be beneficial to evaluate if the method developed in this study could be beneficial to these investigations. Other systems, | | 542-573 | 591-622 |

| beyond coastal environments, could also be evaluated here to increase the application of the work. With these additions, the manuscript would be considerably stronger and of wider appeal to those working in both UAS remote sensing and blue carbon. | | | |
| --- | --- | --- | --- |

| Reviewer Comment | Response | Lines of Change (track change view off) | Lines of Change (track change view on) |
| --- | --- | --- | --- |
| Please enhance the resolution of the figures and the legibility of text within the figures. | All Figures have been amended. To aid this, the previous Figure 3 has now been split into two separate figure: one with just the RGB orthomosaics (Figure 3) and another with the DSM and NDVI orthomosaics (Figure 4). All subsequent Figures have been renumbered. | | |
| Provide a more detailed description of the study sites, including specific information on vegetation species and associations. | We have added a new section (2.2) with more detailed site descriptions. We have also added a new Table (new Table 1) which outlines the established vegetation NVC categories and their associated saltmarsh relationships. This Table has been added in Section 2.5, where we introduce the method underpinning NVC classification. Given that the vegetation associations are a key part of the results of our survey, we feel that it site in Section 2.5 better than it would in our site descriptions. | 112-127 | 123-142 |
| The dataset collected during the field surveys lacks sufficient description. Improve this section by including important information, such as the number of GPS observations conducted, and specify the accuracy of the dGPS measurements. | We have reworked Section 2.3 to make it clearer that every quadrat location (number for each site is stated) has an associated dGPS location. We have added the model of the GPS and the accuracy. | 131-134 | 146-149 |
| The manuscript lacks a comprehensive and detailed | We have added detail to Section 2.8 expanding on the use of the | 265-280 | 309-325 |

| | | | |
|---|---|---|---|
| explanation of the model and field measurements used by the authors to estimate the organic carbon (OC) distribution across the marsh. Clarify and provide additional details. Expand upon the use of the Markov Chain Monte Carlo (MCMC) methodology in the Methods section. | MCMC to estimate carbon storage at Loch Fleet. | | |
| Regarding the statistics concerning soil elevation values corresponding to each vegetation community, please specify the statistical test employed to establish the significance of the analyses. | In the manuscript we don't attempt to statistically define the elevations associated with each saltmarsh community. The elevational trends are useful for determining the reliability of our NVC community classification, but for many of the communities we have insufficient data to statistically determine whether communities fall into significantly different elevation ranges. This is something that would be interesting in the follow-up work suggested in our conclusions – determining the relationship between individual saltmarsh communities and carbon storage using additional remote sensing and ground-truthing techniques. For this study, however, community vegetation is not a significant component of the method (the exception being the elevation of the community in relation to the High Astronomical Tides). | | |

---

## Referee Report (RR1)

The authors addressed all comments carefully which led to a great improvement of the manuscript. Good job!

Although the authors described the classification scheme of NVC system, I am still concerned about the accuracy of vegetation mapping, since it is related to later organic carbon storage estimates.

Line 74, the sentence in the bracket, I don't think it is a good idea to put this sentence here. This could be discussed more and clarified the advantage of method you proposed in the discussion section.

Figure 3, The layout of three quadrat sampling locations seems not good. Could you show the overall study area in the left, and by side show three specific study areas?

Table 1-5, at least the bottom frame line should be added.

---

## Author Response (AR2)

The authors addressed all comments carefully which led to a great improvement of the manuscript. Good job!

We thank the reviewer for taking the time to review the revised manuscript and provide this final set of comments. We believe that all the queries are now resolved, and the manuscript is suitable for publication.

Although the authors described the classification scheme of NVC system, I am still concerned about the accuracy of vegetation mapping, since it is related to later organic carbon storage estimates.

The standard methodology for vegetation survey was used in this study which matches that used by Haynes, 2016)

To provide greater clarity the text has been updated to with more detail on the methodology used.

**Line 97 -** These saltmarshes were last surveyed in 2011 (Haynes, 2016) following the National Vegetation Classification (NVC) scheme approach (Rodwell, 2000).

**Line 132 -** Along each transect several 1 $m^2$ quadrats were placed approximately every 10 m or at abrupt changes in vegetation.

**Line 135 -** At each quadrat, the vegetation plant species were identified, and their percentage coverage was estimated by eye following standard NVC methodology (Rodwell, 2000) as used by Haynes. (2016). Additionally, the mean (n = 5) and maximum (n = 1) vegetation heights were determined with the quadrat following Stewart et al. (2001).

**Line 159 -** Post-processing followed the processes outlined in Fig. 3. Vegetation data was processed into the NVC scheme (Rodwell, 2000) using Modular Analysis of Vegetation Information System (MAVIS). This method uses a United Kingdom reference database for vegetation communities and, using multivariate statistical methods, assigns survey data to an established community based on the community composition (Table 1). Some communities may be classified as a mosaic, being comprised of one or more sub-community, but where this occurs the dominant community is used. Where the prefix "SM" occurs in front of a numerical value, this denotes a saltmarsh vegetation community classified using the NVC approach.

In this manuscript we estimate the OC stocks as illustration of the importance of having accurate up to date areal extent data. Therefore, for the purposes of this manuscript we do not use individual plant community to upscale. Rather for the purposes of this manuscript we calculate the stock based on total marsh extent calculated from Haynes, 2016 vs the areal extent calculated from the UAV approaches.

Line 74, the sentence in the bracket, I don't think it is a good idea to put this sentence here. This could be discussed more and clarified the advantage of method you proposed in the discussion section.

We agree that this should not be the introduction. Section 4.4 discusses this in more detail.

Figure 3, The layout of three quadrat sampling locations seems not good. Could you show the overall study area in the left, and by side show three specific study areas?

We have now combined Figure 3 with Figure 1. For space reasons, we have opted to position the three orthomosaics below the full study area. All other Figures and in-text Figure references have been updated accordingly.

Table 1-5, at least the bottom frame line should be added.

Lines have been added to tables 1 to 5.